# Molecular Phylogenetics and the Evolution of Morphological Complexity in Aytoniaceae (Marchantiophyta)

**DOI:** 10.3390/plants13081053

**Published:** 2024-04-09

**Authors:** You-Liang Xiang, Chao Shen, Wen-Zhang Ma, Rui-Liang Zhu

**Affiliations:** 1School of Life Science, Guizhou Normal University, Huaxi District, Guiyang 550025, China; 2Bryology Laboratory, School of Life Sciences, East China Normal University, 500 Dongchuan Road, Shanghai 200241, China; shenchao_hznx@163.com (C.S.); rlzhu@bio.ecnu.edu.cn (R.-L.Z.); 3Herbarium, Key Laboratory for Plant Diversity and Biogeography of East Asia, Kunming Institute of Botany, Chinese Academy of Sciences, Kunming 650201, China; mawenzhang@mail.kib.ac.cn; 4Tiantong National Station of Forest Ecosystem, Shanghai Key Laboratory for Urban Ecological Processes and Eco-Restoration, East China Normal University, 500 Dongchuan Road, Shanghai 200241, China

**Keywords:** ancestral character reconstructions, *Asterella*, *Asterellopsis*, complex thalloid liverworts, divergence time, new combination

## Abstract

Aytoniaceae are one of the largest families of complex thalloid liverworts (Marchantiopsida), consisting of about 70 species, with most species being distributed in temperate areas. However, the phylogeny and evolution of the morphological character of Aytoniaceae are still poorly understood. Here, we employed two chloroplast loci, specifically, *rbc*L and *trn*L-F, along with a 26S nuclear ribosomal sequence to reconstruct the phylogeny and track the morphological evolution of Aytoniaceae. Our results reveal that Aytoniaceae are monophyletic, and five monophyletic clades were recovered (i.e., *Asterellopsis*-*Cryptomitrium*, *Calasterella*, *Mannia*, *Reboulia*-*Plagiochasma,* and *Asterella*). *Asterella* was divided into five clades (i.e., *Asterella lindenbergiana*, subg. *Saccatae*, subg. *Phragmoblepharis*, subg. *Wallichianae,* and subg. *Asterella*), except for *Asterella palmeri,* which is the sister of *Asterellopsis grollei*. Bayesian molecular clock dating indicates that the five primary clades within Aytoniaceae underwent divergence events in the Cretaceous period. *Asterellopsis* differentiated during the early Upper Cretaceous (c. 84.2 Ma), and *Calasterella* originated from the late Lower Cretaceous (c. 143.0 Ma). The ancestral Aytoniaceae plant is reconstructed as the absence of a pseudoperianth, lacking equatorial apertures, and having both male and female reproductive organs on the main thallus. At present, *Asterellopsis* consists of two species known in Asia and America with the new transfer of *Asterella palmeri* to *Asterellopsis.* A new subgenus, *Asterella* subg. *Lindenbergianae*, is proposed.

## 1. Introduction

Marchantiopsida, commonly referred to as complex thalloid liverworts, represent one of the earliest branches in the evolutionary history of terrestrial plants. They are a class of liverworts, which are small, non-vascular plants belonging to the phylum Marchantiophyta. They are often found in damp or aquatic environments and are characterized by their flattened, ribbon-like, or leafy gametophytes, which usually have air pores and chambers. Aytoniaceae Cavers, the third largest family of Marchantiopsida following Ricciaceae and Marchantiaceae, are recognized as one of the most challenging taxonomic groups within the complex of thalloid liverworts. Traditionally, Aytoniaceae have been thought to have a close relationship with Cleveaceae. However, recent phylogenetic studies based on chloroplast genomes have revealed that Aytoniaceae constitute the second branching node within Marchantiidae [1]. Recent phylogenetic studies have also shown that Aytoniaceae resolve into five major clades, including *Asterella grollei* D.G. Long, which forms a robustly supported clade with *Cryptomitrium*, and *Asterella californica* (Hampe ex Austin) Underw. in a single clade [2]. Based on the phylogenetic and morphological evidence, two new genera, *Asterellopsis* R.L. Zhu and You L. Xiang and *Calasterella* D.G. Long and T.X. Zheng, have recently been proposed in Aytoniaceae [1,3]. Therefore, there are now seven genera in this family. The systematic relationships between genera in Aytoniaceae are becoming clear, but the phylogenetic relationships among subgenera within *Asterella* remain unclear and require further resolution [4,5]. 

*Asterella palmeri* (Austin) Underw. was originally described as “*Fimbriaria palmeri*” Austin and was transferred to *Asterella* P. Beauv. by Underwood in 1895 [6]. The latter treatment has been widely adopted since then [7,8,9]. A recent molecular study of complex thalloid liverworts based on chloroplast genomes and morphological characters showed that *A*. *grollei* should be placed in its own genus, *Asterellopsis* [1]. Therefore, the phylogenetic status of *Asterella palmeri*, *A*. *saccata* (Wahlenb.) A. Evans, and *A*. *muscicola* (Steph.) S.W. Arnell, which are morphologically similar to *Asterellopsis grollei*, should be reassessed. 

Nuclear ribosomal DNA (nrDNA) (26S) and two chloroplast coding regions (*rbc*L and *trn*L-F) with a wide range of phylogenetic significance have been extensively used in molecular phylogenetic studies across various plant groups, especially in Marchantiopsida [2,10,11]. Here, we present the most extensive phylogenetic analysis of Aytoniaceae to date, encompassing samples from all seven genera by using these molecular markers. By expanding taxon and character samplings, our study aims to (1) examine the monophyly of the four subgenera within *Asterella*, (2) estimate the differentiation time of major clades in Aytoniaceae, and (3) reconstruct the phylogeny and morphological evolution of Aytoniaceae.

## 2. Results

### 2.1. Phylogenetic Analyses

The composite dataset comprising 88 taxa contained 3239 aligned nucleotides, with 2607 being constant sites and 466 providing parsimony-informative data. The Aytoniaceae clade is firmly established as monophyletic, with substantial support (MLBS = 100; PP = 1). Within Aytoniaceae, the seven previously identified genera are all strongly supported as monophyletic, except *Asterella palmeri,* which is sister to *Asterellopsis grollei* (MLBS = 98; PP = 1) (Figure 1). The *Cryptomitrium*-*Asterellopsis* clade is sister to the rest of Aytoniaceae (MLBS = 100; PP = 1), followed by *Calasterella*, *Mannia*, *Reboulia*-*Plagiochasma,* and *Asterella*. The genus *Asterella* is divided into five clades (Clade I: subg. *Asterella*, Clade II: subg. *Wallichianae*, Clade III: subg. *Phragmoblepharis*, Clade IV: subg. *Saccatae,* and Clade V: *Asterella lindenbergiana*). *Asterella lindenbergiana* (Clade V) is sister to all of the other subgenera of *Asterella* (MLBS = 88; PP = 1). The backbone branching order within *Asterella* is *Asterella lindenbergiana* (subg. *Saccatae* (subg. *Phragmoblepharis* (subg. *Wallichianae* (subg. *Asterella*)))) (Figure 1).

### 2.2. P-Distances

The observed intergeneric p-distances from all genera of the family Aytoniaceae are 1.5–3.6% in the *rbc*L region, 4.8–11.4% in the *trn*L-F region, and 2.3–4.4% in the 26S region (Table 1). The calculated p-distances for *Asterella palmeri* and *Asterellopsis* differ across the *rbc*L, *trn*L-F, and 26S regions, with values of 2.1%, 8.4%, and 3.2%, respectively, which are lower than divergence among some genera in Aytoniaceae (Table 1). 

### 2.3. Divergence Time Estimations

According to our molecular dating analysis (Figure 2 and Table 1), we estimated the crown age of Aytoniaceae (node 1) to be 145.35 Ma (95% HPD: 118.52–177.46 Ma). The age of the *Cryptomitrium*-*Asterellopsis* clade (node 4) was estimated to be 107.43 Ma (95% HPD: 73.71–145.92 Ma), while those of *Asterellopsis* (node 7) were estimated at 84.13 Ma (95% HPD: 47.75–119.83 Ma). *Calasterella* (node 2) diverged from Aytoniaceae during the late Lower Cretaceous (143.03 Ma (95% HPD: 115.26–174.55 Ma)). The species-rich *Asterella* (node 8) originated during the early Lower Cretaceous (121.73 Ma (95% HPD: 94.63–154.49 Ma)). 

### 2.4. Ancestral State Reconstructions of Complex Traits

Ancestral state reconstructions of the pseudoperianth (Figure 3a) identified the absence of the pseudoperianth as the most likely ancestral condition for Aytoniaceae (*p* = 0.71) and other genera, including *Cryptomitrium* (*p* = 1.00), *Mannia* (*p* = 1.00), *Reboulia* (*p* = 1.00), and *Plagiochasma* (*p* = 1.00). The most probable ancestral state for *Asterella* (*p* = 0.88), *Asterellopsis* (*p* = 0.90), and *Calasterella* (*p* = 0.92) is the presence of a pseudoperianth. In Aytoniaceae, there are three independent absence-to-presence transitions of the pseudoperianth. In *Mannia*, the majority of species maintained the ancestral condition of lacking a pseudoperianth, with a transition from the absence to the presence of the pseudoperianth being inferred only in *Mannia gracilis*, which is known in Asia, Europe, and North America (see Figure 3a).

For spore equatorial apertures, the absence of spore equatorial apertures was inferred as the ancestral condition for the most recent common ancestor of Aytoniaceae (*p* = 0.98). Likewise, the absence of spore equatorial apertures was reconstructed as the ancestral state in other genera, including *Asterella*, *Cryptomitrium*, *Calasterella*, and *Reboulia* (*p* = 1.00), but the most likely ancestral condition for *Asterellopsis* (*p* = 0.83) and *Mannia* (*p* = 0.83) is the presence of spore equatorial apertures (Figure 3b).

Concerning the sexual conditions (Figure 4), the ancestral state reconstruction indicated that main-thallus autoicy was the most likely ancestral condition for Aytoniaceae (*p* = 0.85). Within the clade of *Asterella*, the sexual conditions of most species changed from main-thallus autoicy (ancestral state) to dioicy, female-ventral autoicy, male-ventral autoicy, and female–male-ventral autoicy. In Aytoniaceae, there are three independent transitions from monoicy (autoicy) to dioicy. 

## 3. Discussion

### 3.1. Phylogenetic Relationships within Aytoniaceae

Previous studies have clarified the phylogenetic placement of the family Aytoniaceae based on complete plastid genome data [1]. However, owing to fragmentary sampling in the plastome analyses, the phylogenic positions of genera in Aytoniaceae remained ambiguous. This study, based on sampling that covered all of the genera in Aytoniaceae, suggested that the backbone branching order within Aytoniaceae is (*Asterellopsis*, *Cryptomitrium*), (*Calasterella* (*Mannia* (*Reboulia*, *Plagiochasma* (*Asterella*)))) (Figure 1). Our results support the establishment of two monophyletic genera, *Asterellopsis* and *Calasterella* [1,3]. *Asterella palmeri* is a sister to *Asterellopsis grollei* with high support values (MLBS = 98; PP = 1) and shares some important morphological characteristics with *Asterellopsis grollei,* including female receptacles that are spherical not dehiscent, flap-like involucres with pseudoperianths, and spores with equatorial apertures (Figure 5). Meanwhile, the intergroup average distance between *Asterella palmeri* and *Asterellopsis* (Table 1) is lower than most of its distances to other genera of Aytoniaceae. Based on this morphological evidence, the phylogenetic relationships, and the level of genetic differences, we propose the transfer of *Asterella palmeri* to *Asterellopsis* as *Asterellopsis palmeri*, comb. nov.

Based on sampling that covers all of the sections and subgenera in *Asterella*, *A*. *lindenbergiana* is composed of a single clade (clade V) and is a sister to the other subgenera of *Asterella*. Morphological traits indicate that *A*. *lindenbergiana* stands apart from other subgenera within *Asterella* due to its distinct features, including a free, deeply sinuate-lobed, rounded margin on the involucre, a pinkish tubular pseudoperianth, and dark purple spores characterized by numerous small alveoli on the winged outer surface. *Asterella* sect. *Lindenbergianae* Grolle was proposed within subg. *Phragmoblepharis* by Grolle [12], and it includes only one species, *Asterella lindenbergiana* (Corda ex Nees) Lindb. ex Arnell. 

The significant morphological differences of *A*. *lindenbergiana* and the results of its phylogeny reveal that the erection of a new subgenus, *Lindenbergianae*, is necessary to accommodate this interesting species.

### 3.2. Evolution of the Pseudoperianth, Spore Equatorial Apertures, and Sexual Conditions in Aytoniaceae

Our ancestral state reconstructions inferred the absence of pseudoperianth and spore equatorial apertures as the most likely ancestral traits of Aytoniaceae (Figure 3a,b). The presence of a pseudoperianth has been used as the main characteristic to identify *Asterella* [7,8]. However, the results from the ancestral state reconstructions show that the pseudoperianth is a plesiomorphic feature for the whole of Aytoniaceae ([13] in this study). The pseudoperianths have been lost in *Cryptomitrium*, *Mannia* (except *Mannia gralicis*), *Reboulia*, and *Plagiochasma* but acquired in *Asterella*, *Asterellopsis*, *Calasterella*, and *Mannia gralicis*. There are at least four independent presence-to-absence transitions of the pseudoperianth in Aytoniaceae (Figure 3a). 

Spore equatorial apertures are an overlooked morphological feature that occurs in some Marchantiales, including Aytoniaceae and Ricciaceae. In Aytoniaceae, they are present in *Asterellopsis*, *Mannia*, and *Plagiochasma*. Compared with the absence of spore equatorial apertures, the presence of spore equatorial apertures is an evolutionary trait; there were at least three independent absence-to-presence transitions of spore equatorial apertures in Aytoniaceae (Figure 3b).

To meet environmental challenges, Aytoniaceae plants have diversified their sexual conditions through evolutionary adaptation [8]. Most Aytoniaceae species (about 90%) are monoicous (autoicous), which is quite different from the whole of bryophytes, in which 68% of liverworts and 57% of mosses are dioicous [14]. However, in Aytoniaceae, most Aytoniaceae species (more than 90%) are monoicous (autoicous) (Figure 4). Obviously, sexual conditions are more variable in *Asterella* than in other genera. Sexual conditions changed from ancestral main-thallus autoicy to dioicy, female-ventral autoicy, male-ventral autoicy, and female–male-ventral autoicy in *Asterella* (Figure 4). Ando (1980) suggested monoicy as advanced over dioicy in bryophytes for seven reasons [15]. This direction of evolution is inconsistent with our study in Aytoniaceae. 

### 3.3. Taxonomy

*Asterella* subg. *Lindenbergianae* (Grolle) You L. Xiang and R. L. Zhu, *stat. nov.*

≡*Asterella* sect. *Lindenbergianae* Grolle, Feddes Repertorium 87: 246. 1976.

Type species: *Asterella lindenbergiana* (Corda ex Nees) Lindb. ex Arnell

(≡*Fimbraria lindenbergiana* Corda ex Nees, Naturg. Europ. Leberm. 4: 266, 283. 1838.)

Description: Thalli thick, prostrate, forming patches on mats, with the smell of rotten fish, regularly dichotomously branched, and leathery. The dorsal epidermis is delicate to firm, with scattered oil cells and slightly to moderately elevated simple pores. Assimilatory tissue is well-developed, loose, and there is a 2–4-layered air chamber without free secondary filaments. Rhizoids are smooth and pegged. Ventral scales are in reddish-purple hues, shaped as rounded triangles to ovals, positioned on both sides of the midrib, and slightly overlapping. Each scale has 1–2 lanceolate to oblong-lanceolate appendages, varying in color from hyaline to reddish. Monoecious. Antheridia emerge dorsally on the primary thallus, just after the female receptacle. The stalk of the receptacle has a solitary rhizoidal furrow. The involucre edge is free, deeply sinuate-lobed, and rounded. The pseudoperianth is pinkish, tubular, and terminates in 12–16 reddish-brown lanceolate segments fused at the apex. Spores are dark purple, winged, and 70–90 μm in diameter, with a surface adorned with numerous small alveoli, accompanied by an indistinct trilete marking. Elaters are in a 1–2-spiral.

Range: Monospecific subgenus known in America, Austria, Canada, Finland, France, Germany, Italy, Mexico, Norway, Poland, Romania, Slovakia, Slovenia, Spain, Sweden, Switzerland, and Ukraine [4,16]. 

*Asterellopsis palmeri* (Austin) You L. Xiang and R. L. Zhu, *comb. nov.* (Figure 5D–F)

≡*Fimbraria palmeri* Austin, Bull. Torrey Bot. Club 6 (7): 47. 1875.

≡*Asterella palmeri* (Austin) Underw., Bot. Gaz. 20 (2): 63. 1895. 

Type: America, Lower California, Guadalupe Island, 1875, *Palmer 119* (NY).

Description: Thalli thick, prostrate, forming patches on mats, dichotomously branching, with purplish margins; intercalary lateral branches are infrequent. Thalli are 5–10 mm × 2–4 mm in size; the epidermis surface is smooth, with thin-walled cells and minute trigones, without oil cells; air pores are encircled by 5–6 cells characterized by slightly thickened radial walls; ventral scales are overlapped, deep purple in color, and scattered with oil cells; appendages are usually 1–2 in number, displaying hues of purplish or hyaline, with a tapering or acuminate shape, and they are often adorned with minor marginal teeth that slightly protrude beyond the edges of the thallus. Monoecious; androecia are at the posterior base of female stalks; the female receptacle terminal is on the thalli; stalks are naked with a single rhizoidal furrow; carpocephala are ovoid or conical, with 3–4 lobes directed downward; the pseudoperianths are white and divided into 8–10 segments with attached tips. Capsules are spheroid; spores are dark brown and 60–80 µm in diameter; elaters in two spirals.

Range: Known only in California, New Mexico, and Guadalupe Island, Mexico [16].

## 4. Materials and Methods

### 4.1. Taxon Sampling

In the present study, we sampled all seven genera within Aytoniaceae, as well as each subgenus and section of *Asterella*. Based on previous molecular phylogenetic studies of the family, *Dumortiera hirsuta* (Sw.) Nees, *Targionia hypophylla* L., and *Wiesnerella denudata* (Mitt.) Steph. were selected as outgroups. Eighty-eight accessions were included in the phylogenetic analyses. The ingroup included 83 accessions of 34 taxa from Aytoniaceae (Appendix A). Seventy-three accessions of Aytoniaceae were newly generated, while the remaining sequences were obtained from GenBank. Detailed information on taxa and vouchers is listed in Appendix A.

### 4.2. Morphological Study

The field images were captured with a Sony ILCE-6000 digital camera (Sony, Beijing, China). SEM studies were performed on a Hitachi S-4800 Scanning Electron Microscope as previously described by Xiang et al. [17] and Xiang & Zhu [18].

### 4.3. DNA Extraction and Sequencing

Plant tissue isolation and total DNA extraction were carried out according to procedures outlined by Xiang and Zhu [19]. Total genomic DNA was obtained utilizing the DNAeasy plant mini kits (Qiagen, Hilden, Germany). The sequencing of one nuclear ribosomal 26S region alongside two chloroplast regions (*rbc*L and *trn*L-F) was carried out by Jie Li Biology Company, Shanghai, China.

### 4.4. Phylogenetic Analyses

PhyDE version 0.9971 [20] was employed for the assembly of all sequences, while Mafft version 7.311 [21] was utilized for alignment purposes. Ambiguous alignment regions were manually trimmed, and missing data were coded accordingly. We employed both the maximum likelihood (ML) and Bayesian inference (BI) methods for conducting phylogenetic analyses. The individual DNA loci, as well as the combined dataset, initially underwent separate analyses. Visual comparisons were then conducted to identify and assess potential incongruences in the topology. Since there were no conflicting nodes observed in the trees, the datasets were merged. ML analyses were conducted using IQtree version 2.0.6 [22] with 1000 sampling repetitions. ModelFinder [23,24] determined the best-fitting substitution models (TN + F + I for *rbc*L, TIM + F + G4 for *trn*L-F, and TIM3 + F + I + G4 for 26S) based on the Bayesian information criterion (BIC). BI was performed with MrBayes version [25] via the CIPRES Science Gateway website [26]. The nucleotide substitution models (GTR + I + G) were selected as the best-fit models for the *rbc*L, *trn*L-F, and 26S partitions according to the Akaike information criterion (AICc). MCMC runs with two independent chains starting from random trees and default priors lasted 2,000,000 generations, with tree sampling every 1000 generations. Convergence was assessed when the average standard deviation of split frequencies (ASDF) reached 0.01 or less. Posterior tree distributions were summarized via a majority-rule consensus (>50%) after discarding the initial 25% of samples as a burn-in.

### 4.5. P-Distances

The intergeneric variability (p-distances) for each DNA locus (*rbc*L, *trn*L-F, and 26S) within Aytoniaceae was computed using Mega version 6.06 [27] while employing the pairwise deletion option to account for gap counts.

### 4.6. Divergence Time Estimation

Due to the limited representation of Aytoniaceae in the fossil record, we can rely with confidence on the previously estimated divergence time reported by Flores [28]: 150.08 Ma (95% HPD: 119.31–178.31 Ma) for the stem age of Aytoniaceae. The MCMC simulations were conducted for 10^7^ generations, with sampling occurring every 1000 generations. We removed the initial 10% of trees as a burn-in, and the convergence of chains was evaluated by examining the output log file using Tracer version 1.7.1 [29]. The MCC chronogram, representing the maximum clade credibility, was graphically presented using Figtree version 1.4.2 (http://tree.bio.ed.ac.uk/software/figtree/, accessed on 11 May 2015).

### 4.7. Ancestral State Reconstructions of Complex Traits

In this research, we chose three significant taxonomic traits (specifically, the pseudoperianth, spore equatorial apertures, and sexual conditions) to reconstruct ancestral states. Data regarding these characteristics were acquired from both sources in the literature and our field observations [8,30]. Subsequently, we employed the BBM method within RASP version 4.2 [31] to conduct ancestral state reconstructions for the pseudoperianth, spore equatorial apertures, and sexual conditions while utilizing the MCC tree derived from BEAST. We tracked the evolutionary history of all three traits within Aytoniaceae: the presence or absence of the pseudoperianth (A, absent; B, present), the presence or absence of spore equatorial apertures (A, absent; B, present), and sexual conditions (A, dioicy; B, main-thallus autoicy; C, female-ventral autoicy; D, male-ventral autoicy; E, female-male-ventral autoicy). Following D. Long’s revision of the genus *Asterella* [8] with our modifications, the sexual conditions recognized in Aytoniaceae were considered as follows: Dorsal-autoicy, par-autoicy, and terminal autoicy were placed in main-thallus autoicy for both antheridia and archegoniophores borne on the main branches of the thallus. A BBM analysis was conducted for 50,000 generations using the fixed-state frequencies model (Jukes–Cantor) with uniform rate variation among sites.

## 5. Conclusions

Our results demonstrate that Aytoniaceae are monophyletic, and five distinct clades were identified: *Asterellopsis*-*Cryptomitrium*, *Calasterella*, *Mannia*, *Reboulia*-*Plagiochasma*, and *Asterella*. Notably, *Asterella* is further divided into five subclades, including *Asterella lindenbergiana* and the newly proposed subgenus, *Asterella* subg. *Lindenbergianae*. Molecular dating analyses indicate that the major clades of Aytoniaceae originated during the Cretaceous period, with *Asterellopsis* emerging during the early Upper Cretaceous (c. 84.2 Ma) and *Calasterella* originating from the late Lower Cretaceous (c. 143.0 Ma). Furthermore, our study provides insights into the morphological characteristics of ancestral Aytoniaceae plants, including the absence of a pseudoperianth, the lack of equatorial apertures, and the presence of both male and female reproductive organs on the main thallus. Additionally, we propose the transfer of *Asterella palmeri* to *Asterellopsis*, expanding our understanding of the diversity within this genus.

Our study offers a foundational framework for future investigations, which may delve deeper into additional morphological characteristics and incorporate fossil evidence to elucidate the origin and evolutionary trajectory of Aytoniaceae, as well as other bryophyte groups.

## Figures and Tables

**Figure 1 plants-13-01053-f001:**
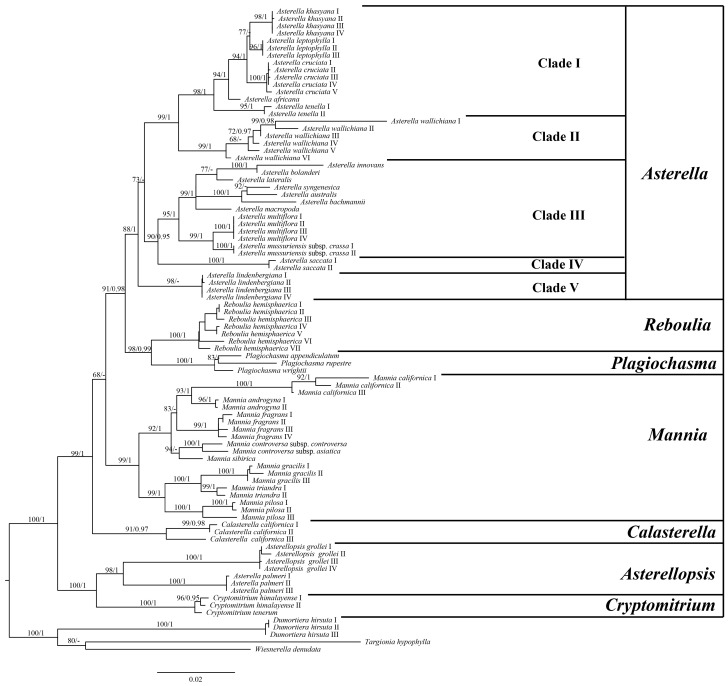
Phylogeny of Aytoniaceae inferred from the combined dataset (*rbc*L, *trn*L-F, and 26S). The topology derived from the best-scoring ML tree in IQtree is shown. ML bootstrap values of BS ≥ 80 and Bayesian posterior probabilities values of PP ≥ 0.95 are shown on the left and at right, respectively.

**Figure 2 plants-13-01053-f002:**
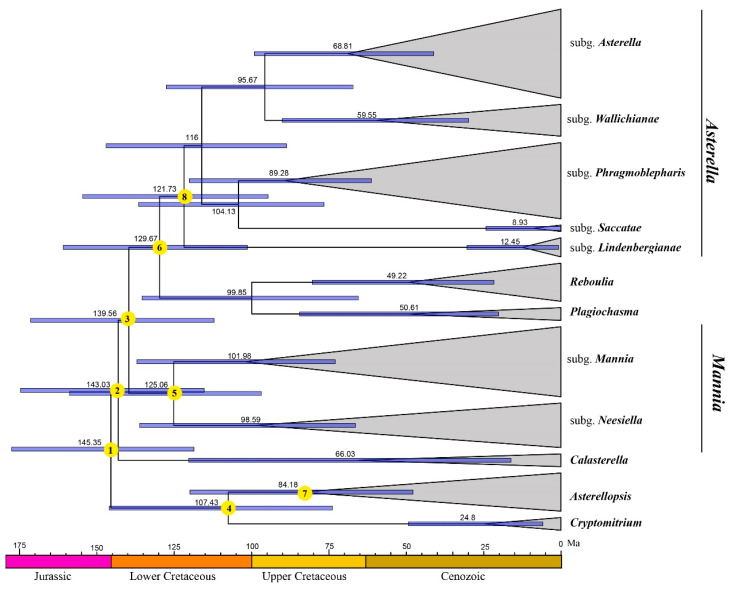
BEAST-derived chronogram of Aytoniaceae based on the combined dataset (*rbc*L, *trn*L-F, and 26S) under a relaxed uncorrelated log-normal clock. The 95% highest posterior density (HPD) intervals around node ages (in million years ago, Ma) and mean relative probabilities of ancestral areas for nodes 1–8 are detailed in Table 2.

**Figure 3 plants-13-01053-f003:**
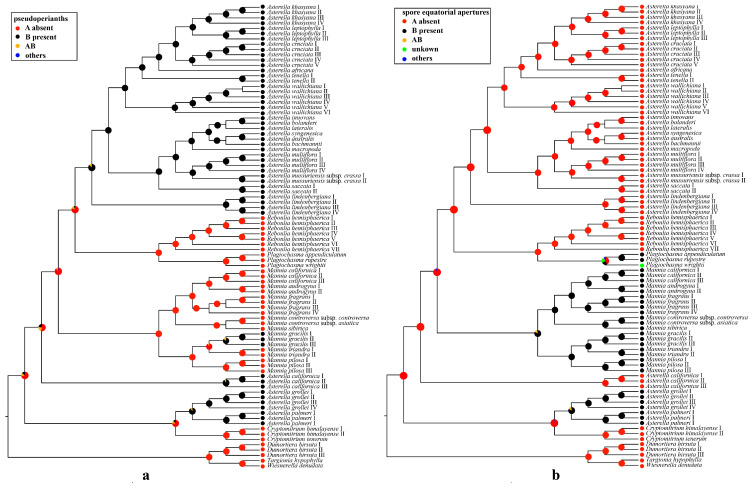
Ancestral state reconstructions for (**a**) pseudoperianth, and (**b**) spore equatorial apertures in Aytoniaceae based on a Bayesian binary MCMC (BBM) analysis in RASP. The corresponding color keys identify extant possible ancestral character states. Pie diagrams at internal nodes indicate the relative probabilities for each alternative state.

**Figure 4 plants-13-01053-f004:**
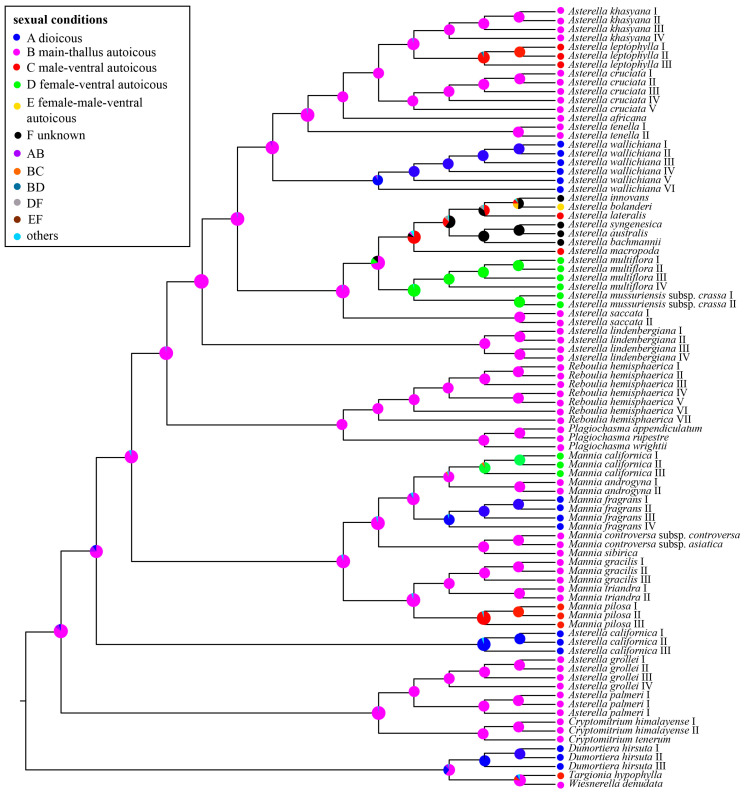
Ancestral state reconstructions for sexual conditions in Aytoniaceae based on a Bayesian binary MCMC (BBM) analysis in RASP. The corresponding color keys identify extant possible ancestral character states. Pie diagrams at internal nodes indicate the relative probabilities for each alternative state.

**Figure 5 plants-13-01053-f005:**
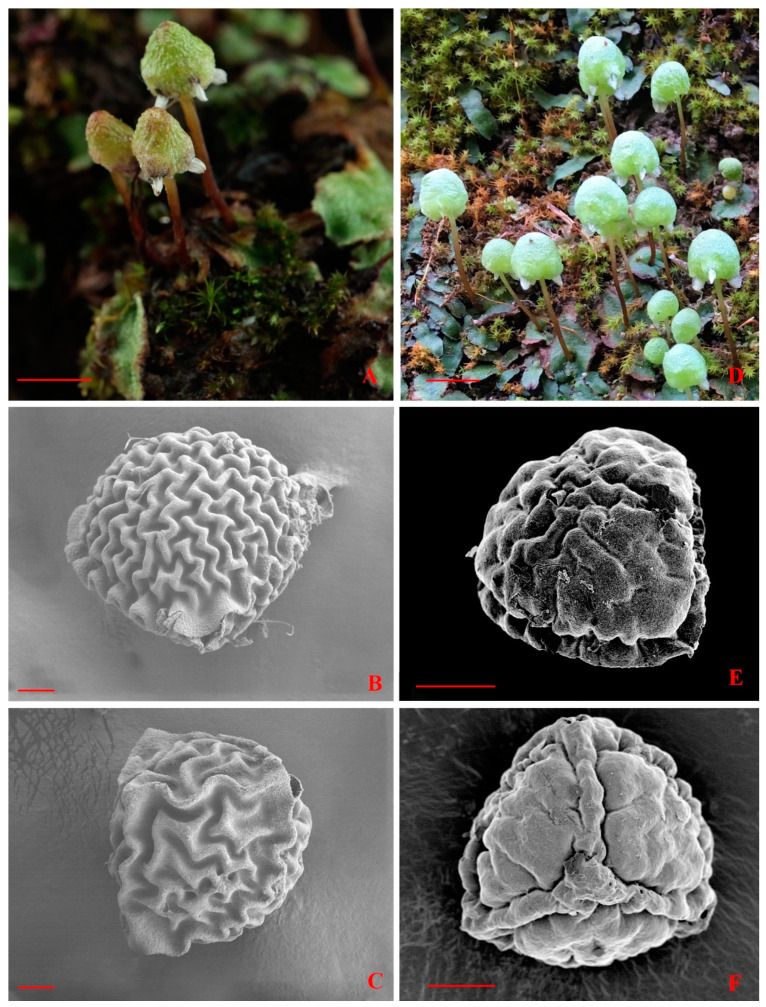
*Asterellopsis grollei* (D. G. Long) R. L. Zhu and You L. Xiang. (**A**) Habit in the field with female receptacles. (**B**,**C**) SEM micrographs of spores, (**B**) distal view, and (**C**) proximal view. All are from *R. L. Zhu 20160815-18A* (HSNU). *Asterellopsis palmeri* (Austin) You L. Xiang and R. L. Zhu. (**D**) Habit in the field with female receptacles. (**E**,**F**) SEM micrographs of spores, (**E**) distal view, and (**F**) proximal view. All are from *Ma 17-9030* (KUN). Scale bars: (**A**,**D**) 5 mm; (**B**,**C**,**E**,**F**) 20 μm.

**Table 1 plants-13-01053-t001:** The values of the intergeneric p-distances of the *rbc*L, *trn*L, and 26S regions (%) for Aytoniaceae.

No. and Taxon	1	2	3	4	5	6	7	8
1. *Asterella*	–	–	–	–	–	–	–	–
2. *Asterella palmeri*	3.5/9.8/3.5	–	–	–	–	–	–	–
3. *Asterellopsis*	3.4/10.6/3.7	2.1/8.4/3.2	–	–	–	–	–	–
4. *Calasterella*	3.3/5.6/4.2	2.6/5.4/4.4	2.4/7.0/4.1	–	–	–	–	–
5. *Cryptomitrium*	3.5/8.9/4.0	3.0/7.0/2.4	2.7/10.3/3.4	3.1/6.2/4.0	–	–	–	–
6. *Mannia*	2.6/7.2/3.6	3.2/9.0/3.7	3.1/10.5/4.3	2.8/5.0/4.0	3.1/8.9/3.9	–	–	–
7. *Plagiochasma*	2.4/7.9/3.9	3.6/9.9/4.0	3.1/11.3/4.3	3.1/6.0/3.9	3.4/7.9/3.4	2.5/7.5/3.3	–	–
8. *Reboulia*	2.0/7.2/2.6	3.1/10.3/4.4	3.1/11.4/4.0	2.8/4.8/3.4	3.3/9.1/3.2	2.2/7.0/3.5	1.5/6.0/2.3	–

**Table 2 plants-13-01053-t002:** BEAST-derived mean node ages and their 95% HPD intervals (in millions of years ago, Ma) for nodes of interest defined in the MCC chronogram for Aytoniaceae.

Nodes	Taxon	Mean Age (Ma)	95% HPD Intervals (Ma)
1	Aytoniaceae	145.35	118.52–177.46
2	-	143.03	115.26–174.55
3	-	139.56	112.05–171.32
4	-	107.43	73.71–145.92
5	*Mannia*	125.06	96.77–158.76
6	-	129.67	101.27–160.76
7	*Asterellopsis*	84.18	47.75–119.83
8	*Asterella*	121.73	94.63–154.49

## Data Availability

The original contributions presented in this study are included in the Appendix A; further inquiries can be directed to the corresponding author.

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
