# Peer review of "Molecular Phylogenetics and the Evolution of Morphological Complexity in Aytoniaceae (Marchantiophyta)"

_plants, 2024, doi:10.3390/plants13081053_

Round 1
Reviewer 1 Report
Comments and Suggestions for Authors
Review report for plants manuscript
General comments: -
This study is very interesting and has a scientific topic with a great impact on the field. The manuscript will be suitable for publication after taking care of the following minor comments.
Detailed comments:
The English language and writing style is fine needs some minor check spelling and grammar check.
Abstract:
This section is well written
Keywords:
-The keywords has been chosen very carefully and accurately.
Introduction:
-This section needs to be elongated and enriched with more background about this topic.
Materials and Methods
It is ok and adequate
Results:
The results are very interesting and well presented.
Discussion:
This section is ok but it would be better if the results and discussion were combined in one section.
References
This section is poor and doesn’t have enough citations . Please add needs more related and UpToDate citations.
Comments on the Quality of English Language
Some minor modifications are required
Author Response
Thank you very much for taking the time to review this manuscript. Please find the detailed responses below and the corresponding revisions/corrections highlighted/in track changes in the re-submitted files.
We agree with some of the comments. We have revised the introduction part of the article and add some references to some extent. And we have observed that our research background is somewhat concise, thus we have already opted to switch the article type to Communication.
Reviewer 2 Report
Comments and Suggestions for Authors
The paper is well structured and I hope the authors will be able to deepen their phylogenetic research in the field of Marchantiopsida
Author Response
Thank you for your feedback. We greatly appreciate your encouragement to delve deeper into phylogenetic research within the field of Marchantiopsida. We will certainly consider your suggestion moving forward.
Reviewer 3 Report
Comments and Suggestions for Authors
Overall, a very clear and well-written manuscript. My only query is why Neesiella is recognized as a subgenus of Mannia rather than a separate genus. Please provide an explanation.
Comments on the Quality of English LanguageFamily names (e.g., Aytoniaceae) are plural nouns and take plural verbs. Therefore, it should be Aytoniaceae are, NOT Aytoniaceae is. See lines 3, 18, 33, 35, 40, 107. In line 33, remove the word "of" between complex and thalloid. In line 70, add a space between -nine and accessions. Line 82, remove \ at front. Line 109, change the semicolon (;) to a period (.) after Aytoniaceae.
Reviewer 4 Report
Comments and Suggestions for Authors
Line 34-35: Please clarify the term “Marchantiopsida” Marchantiopsida is a class of liverworts within the phylum Marchantiophyta. Because the species in this class are known as complex thalloid liverworts. Later moves toward the Family Aytoniaceae.
Line 46. Grammatical mistake. Start a new sentence from “However…”
Line 58: “DNA (nrDNA) (26S), two chloroplast (cp) coding (rbcL and trnL-F)” Why did the authors choose the two chloroplast genes? Provide their wide range of phylogenetic significance first and then move toward a specific topic.
Line 65-66: “Our sampling contained all seven genera within Aytoniaceae, and all subgenera and sections of Asterella” How did the authors collect the species? From where do the authors collect the species? How do they preserve the samples? Are the samples deposited in the herbarium or not? How do they identify the species? Does any database follow for the verification of the species names?
In the methodology section, how do the authors analyze the spore? Please provide separate headings such as “Microscopic study” and then provide details of how the spores were collected, and separated, and which chemical was used. Which microscope along with the model name? where perform microscopy? and how to make the samples for the microscopic analysis.
Line 74: “Methods were used as in Xiang et al. [10] and Xiang & Zhu [11].” Not clear please clarify this statement first and then cite the citations.
Line 76: Revise this sentence.
Line 82: Typographic mistake.
In the result section the spore description was not presented clearly, please provide a separate heading for the spore and species morphology and use the correct terminology. Such as alete, trilete based on the number of apertures of the spore. Likewise for the perine structures use correct terminology such as psilate, regulate or scabrate. What about the spore size? For example (23 x 22 µm). After mentioning both the macro and micromorphological results in the results section, then discuss and compare these findings with the literature in the discussion section.
Line 69. The authors mentioned 34 species but how many spores the authors investigated using a scanning electron microscope?
Please provide a separate heading for the conclusion and conclude your findings. At the end of the conclusion please provide a future recommendation as well.
Comments on the Quality of English Language
Minor editing of English language required
Round 2
Reviewer 4 Report
Comments and Suggestions for Authors
The authors revised the manuscript well and therefore, recommended for publication in its present form.
Comments on the Quality of English LanguageMinor editing of English language required